# Benefit of urban greenness on patients after an ischaemic stroke: mortality or recurrence? A registry-based cohort study

Raphaël Anxionnat[1,2], Nadine Bernard[3], Anne-Sophie Mariet [4,5], Sophie Pujol[2,3], Anne-Laure Parmentier[2,3], Kadiatou Diallo[2], Hélène Houot[6], Théophile Pierre[2], Yannick Béjot[7], Frédéric Mauny[2,3]*

**1** CHU Besançon, Service de médecine pédiatrique, Besançon, France, **2** CHU Besançon, Inserm CIC, uMETh, Besançon, France, **3** Université Marie et Louis Pasteur, CNRS, Chrono-environnement (UMR 6249), Besançon, France, **4** Université Bourgogne Europe, CHU Dijon Bourgogne, Service Biostatistiques et Information Médicale, Dijon, France, **5** Université Bourgogne Europe, CHU Dijon Bourgogne, Centre d'investigation clinique, Module Épidémiologie Clinique, Inserm, CIC, Dijon, France, **6** Université Marie et Louis Pasteur, CNRS, THéMA (UMR 6049), Besançon, France, **7** Université Bourgogne Europe, CHU Dijon Bourgogne, Service de Neurologie, Dijon Stroke Registry, UR7460, Dijon, France

* frederic.mauny@univ-fcomte.fr

## Abstract

### Background

Whether living environment may influence outcome of stroke survivors remains to be elucidated.

### Aim

This registry-based cohort study aimed to assess the relationship between urban greenness around the residence and one-year death or recurrence after a first-ever ischaemic stroke.

### Methods

Patients with a first-ever ischaemic stroke who directly returned home were identified from the population-based registry of Dijon, France. For each patient, after geolocation of residential building, two greenness indices were calculated: the distance by road and pedestrian networks to the nearest public green space, and the area of green spaces within radii of 100 and 400 metres. Atmospheric $NO_2$ and $PM_{10}$ outdoor concentrations around the residence and deprivation index were assessed.

### Results

During the 2005–2008 study period, 360 patients were identified and included (median age: 75 years-old (IQR: 63–83), 56% women). Fifteen died and 17 had recurrent stroke during the one year of follow-up. In adjusted models, the distance

**Data availability statement:** The data underlying this study contain potentially identifying and sensitive patient information. In particular, the use of geocoded residential data in a relatively small cohort could allow indirect identification of participants. Public sharing of these data is restricted by French regulations. Data are therefore available upon reasonable request sent to: Registre Dijonnais des AVC - Direction de la recherche clinique et de l'innovation - CHU Dijon Bourgogne 1 Boulevard Jeanne d'Arc 21079 DIJON Cedex, France recherche @chu-dijon.fr.

**Funding:** NO2 pollution exposure assessment was supported by the Agence de l'Environnement et de la Maîtrise de l'Énergie (ADEME) [1217C0065] as part of the National Environmental Research and Occupational Health Program 2012 from the French Agency for Food, Environmental and Occupational Health and Safety (ANSES). PM10 pollution exposure assessment was supported by the Fondation de France (Engagement 00089823). The Dijon Stroke Registry was funded by Santé Publique France (French Institute for Public Health Surveillance), Institut national de la santé et de la recherche médicale (INSERM), and CHU Dijon Bourgogne. The funders had no role in study design, data collection and analysis, decision to publish, or preparation of the manuscript.

**Competing interests:** Yannick Béjot reports personal fees from BMS, Pfizer, Medtronic, Amgen, Servier, NovoNordisk, Novartis, outside the submitted work. Other authors: none. This does not alter our adherence to PLOS ONE policies on sharing data and materials.

between public green spaces and patients' residence was associated with stroke recurrence or death (HR = 1.26, 95% CI: 1.08–1.48, $P < 0.01$, for each 100 metre section of city network). In age-stratified analysis, this association remained significant only in patients aged 65–79 years (HR: 1.37, 95% CI: 1.10–1.71, $P < 0.01$). When considering separately stroke recurrence and death, this association remained significant for recurrence (HR = 1.30, 95% CI: 1.07–1.58, $P < 0.01$) but not for death (HR = 1.17, 95% CI: 0.89–1.52).

## Conclusion

This study highlighted a beneficial influence of greenness on post-stroke recurrence in an urban area. These results indicate that urban planning policy could impact secondary prevention.

## Introduction

Despite improvements in both prevention and acute management over the past decades, the burden of ischaemic stroke remains a major issue worldwide that will magnify in the coming years because of the ageing population and demographic transition [1], with survivors facing a substantial risk of recurrence and premature death. In addition to traditional vascular risk factors, increasing attention has been directed toward the role of environmental determinants of stroke. In a context of urbanization of lifestyles, with more than half of the world population living in urban areas, a substantial body of literature has pointed out an association between short- or long-term exposure to high ambient air pollution levels and ischaemic stroke occurrence [2,3]. Conversely, the impact of residential green space exposure on stroke burden has been less documented [4]. A few studies with heterogeneous design and population setting showed a protective association between green space exposure and either stroke incidence [5–11] or stroke-related mortality [12–15]. Although largely hypothetical and multifactorial, proposed mechanisms underlying this association include improvements in cardiovascular and metabolic regulation (e.g., decreased heart rate and diastolic blood pressure [16]), reductions in primary atmospheric pollutants and noise, enhanced immune function [17,12], and promotion of physical and social activity [18]. The role of the human microbiota in autoimmune or inflammatory disorders has gradually been established [19,20], and its potential modulation through contact with green spaces has been suggested as a contributing factor [12].

Furthermore, a very limited number of studies focused on stroke survivors specifically, despite their heightened vulnerability to environmental influences in the context of secondary prevention. Hence, their findings suggested that green space exposure may result in a reduced risk of ischaemic stroke severity [21] or post-stroke disability [22].

Surprisingly, to the best of our knowledge, the risk of recurrence of ischaemic stroke in relation to proximity and accessibility to green spaces has not been studied.

To address this gap, this population-based study aimed to analyse the relationship between urban green spaces around the place of residence and one-year outcomes

after a first-ever ischaemic stroke. We hypothesized that greater accessibility to urban greenness would be associated with a reduced risk of death and/or recurrence, independent of clinical, socioeconomic, and environmental confounders.

## Methods

### Study population

This study was based on data obtained from the Dijon Stroke Registry, an ongoing population-based registry [23] that complies with the defined criteria for conducting high-quality incidence stroke studies [24] and the guidelines for the reporting of incidence and prevalence studies in neuroepidemiology according to the Standards of Reporting of Neuro-logical Disorders [25]. The methodology of the Dijon Stroke Registry has been detailed elsewhere [26]. Briefly, multiple overlapping sources of information are used to ensure exhaustive case collection of both hospitalised and nonhos-pitalised cases of stroke and transient ischaemic attack (TIA) among residents of the city of Dijon, eastern France (approximately 156,000 inhabitants currently). All identified cases were adjudicated by senior neurologists trained in stroke assessment according to the World Health Organisation diagnostic criteria [27]. For the present analysis, we only considered patients over 18 years old with first-ever ischaemic stroke who were registered between 01/01/2005 and 31/12/2008, for whom a valid postal address was available, and who were discharged home after hospitalisation or treated as outpatients.

### Outcomes

The primary composite outcome was all-cause death or stroke recurrence at one year. Survival time was calculated from the date of the first stroke to the date of the first event occurrence (death or recurrence). Patients without any occurrence were censored at one year of follow-up. The secondary outcomes were all-cause death and stroke recurrence, which were considered separately.

### Data collected

The following information was collected at registration: sex, age, residence address, and vascular risk factors. Stroke severity at onset was quantified using the National Institutes of Health Stroke Scale (NIHSS) score obtained at the first clinical examination or estimated retrospectively based on medical records and charts, as previously validated in the literature [28]. The aetiological classification of ischaemic stroke patients was performed by a stroke-trained neurologist investigator of the Dijon Stroke Registry based on medical records, including complementary exams performed during the diagnostic workup of ischaemic stroke, as follows: large artery atheroma, cardioembolic ischaemic stroke, lacunar ischaemic stroke due to small vessel disease, and ischaemic stroke from other or unidentified causes. Post-stroke dis-ability was assessed via the modified Rankin scale (mRS) score at discharge from the acute care ward or immediately after stroke for outpatients [29]. Vital status data were collected by regularly checking death certificates obtained from the French national database.

We assessed the global accessibility of urban greenness via two indices: public greenness proximity and greenness density. The degree of public greenness proximity was calculated using a map of public green spaces provided by the mapping department of the Dijon metropolitan area (*Grand Dijon*). After the residential building was geolocated, it was quantified via spatial queries to calculate the distance (by road and pedestrian networks) between the place of residence of each participant and the nearest public green space. The greenness density was calculated via high-resolution satellite images (ORTHO-SAT SPOT) from the French National Geographic Institute to compute the modified soil-adjusted vegeta-tion index (MSAVI) [30,31]. Compared to the normalised difference vegetation index (NDVI), MSAVI detects small vege-tation patches [32] and is better adapted to urban green spaces [31]. This index was computed for the area of the *walking* and *living neighborhood* [33,34]. For each patient, we defined these areas using a circular buffer around the residence. According to Forsyth et al [35] and our previous studies [36,37], the area and percentage of green spaces were calculated

within radii of 100 and 400 metres. The indices were assigned to each patient through their residential address at the date of the first stroke.

Outdoor air pollution exposure was assessed via emission/diffusion models. The method has been previously described [38,39]. Briefly, models were implemented by the regionally approved Air Quality Monitoring Association *ATMO-Bourgogne Franche-Comté* via Circul'Air® and ADMS-Urban® software. Two pollutants were considered: nitrogen dioxide ($NO_2$) and fine particles with a diameter ≤ 10 mm (particulate matter or $PM_{10}$). Air pollution models were validated via seasonal field measurements conducted by air quality monitoring agencies. To adjust for seasonal variations, monthly air pollutant concentrations were calculated. Maps of monthly concentrations expressed in micrograms per cubic metre ($\mu g/m^3$) were available for each pollutant from January 2004 to December 2009, with a 2-metre per 2-metre resolution. For each patient, $NO_2$ and $PM_{10}$ exposure were quantified monthly in the immediate vicinity of the residential building (*i.e.*, the average level within a 50 m radius of the home).

The socioeconomic level of the neighbourhood was estimated via an ecological index of disadvantage with a European vocation, the French DEPrivation index (FDEP). This index is based on four variables: the median income per consumption unit in the household, the percentage of high school graduates in the population aged over 15, the percentage of manual workers in the working population, and the unemployment rate [40]. The statistical unit was the IRIS (*Îlots Regroupés pour l'Information Statistique*), a geographical unit currently used by the French National Institute of Statistics and Economic Studies (INSEE) for population censuses, with approximately 2,000 individuals with relatively homogeneous social characteristics.

## Statistical analysis

Descriptive data are presented as medians and interquartile ranges (IQRs) or means and standard deviations (SDs) for continuous variables or as frequencies and percentages for categorical variables. Multivariable analysis was performed with a Cox survival-type model, with time-varying variables for adjustment on atmospheric $NO_2$ and $PM_{10}$ exposure. Hazard ratios (HRs) and 95% confidence intervals (CIs) related to public greenness proximity were expressed for an increase in distance of 100 metres. The HR and 95% CI related to the greenness density within the 100- and 400-m radii were expressed for increases in the green space area of 1,000 and 10,000 $m^2$, respectively. The analyses of the primary and secondary outcomes were adjusted for sex, age, FDEP, mRS score, and previous 30-day mean atmospheric pollutant concentrations. The significance threshold was set to 0.05. For all analyses, the hypotheses of proportional risks and the absence of marginal subjects, specific to the Cox model, were respected.

We performed a sensitivity analysis by stratifying age groups (<65, 65–79, and ≥80 years old) on the primary outcome and both public greenness proximity and greenness density.

All analyses were performed with R software version 4.1.0 (R Development Core Team, 2005), and the package *survival* was used for survival analyses, including time-varying exposure.

## Ethics

This study is a retrospective analysis of data from the Dijon Stroke Registry, whose accreditation is regularly updated by national scientific or ethics boards: the French National Committee of Registers, the French Data Protection Authority (CNIL, number DR-2017–287), and the French Institute for Public Health Surveillance (Santé Publique France). According to the French legislation boards, the need for written patient consent was waived.

## Results

Over the study period, among the 896 stroke cases recorded in the Dijon Stroke Registry, 360 patients with a first-ever ischaemic stroke were considered. After one year of follow-up, 30 subjects experienced a composite event (13 deaths and

17 recurrences). Among these subjects, two experienced a recurrence and died before the end of the one-year follow up. Regarding separate death and recurrence events, there were 15 and 17 deaths and recurrences.

The sociodemographic, clinical, and environmental characteristics of the patients are described in Table 1. Patients who presented a subsequent event (death or recurrence) were older (median age 81 years [IQR 74–87] vs. 74 years [IQR 63–82], p = 0.002) and had higher post-stroke mRS scores (median 2 [IQR 1–3] vs. 1 [IQR 0–2], p < 0.001). Other characteristics were broadly similar between the subjects. The nearest public green space (public greenness proximity) was located at 309 m on average, and the mean green space areas within 100 and 400 m of the patients' residences (greenness density) were 2,900 and 51,000 m², which corresponded to proportions of 9% and 10%, respectively. The primary outcome (stroke recurrence or death) was associated with public greenness proximity (shown in Table 2). Hence, for each increase of 100 metres, the hazard ratios were 1.17 (95% CI: 1.01–1.37) after adjustment for sex and age (Model 2) and 1.26 (95% CI: 1.08–1.48) after additional adjustment for post-stroke mRS score, FDEP and $PM_{10}$ concentrations (Model 5). No significant association was observed when the greenness density was considered, regardless of the area considered (within a radius of 100 or 400 metres around the residence) (Table 2).

The results of the secondary outcome analyses are reported in Table 3. Whatever the model considered, there was a significant association between stroke recurrence and public greenness proximity. In addition, a higher greenness density, regardless of the area considered, was associated with a reduced risk of stroke recurrence. Conversely, no significant association was observed with death.

The results of the sensitivity analyses (according to age groups) are presented in Table 4. The highest HR related to public greenness proximity was observed for the 65–79-year-old age group. The greenness density did not reveal any association, regardless of the area considered.

## Discussion

This study demonstrated an association between a greater distance from urban public green spaces to the subjects' residence and death and/or recurrence at one year in patients who returned home after a first-ever ischaemic stroke. This association was driven mainly by a higher risk of stroke recurrence and was particularly pronounced in patients aged 65–79 years.

This analysis was performed on data obtained from a population-based registry that provided exhaustive case collection. The patients were living in a middle-sized city, similar to the living conditions of 54% of the European population [41]. In the present analysis, we only considered patients who returned home after a stroke to avoid inaccurate assessment of environmental exposure by using an incorrect residential location (i.e., hospital, rehabilitation centre, etc.). Environmental exposures were directly assessed from the subjects' geocoded addresses via an entirely reproducible process, and special attention was given to identifying the actual home building at the date of the first stroke instead of the last address recorded in the hospital information system. The modelling of exposure allowed a repeatable assessment, irrespective of the vital status of the patients.

Adjustment methods in multivariable analyses take into account the main potential confounding factors, such as sex, age, post-stroke mRS score, or FDEP. The relationship between socioeconomic status and accessibility to public green spaces could be considered complex. However, adjusting for FDEP did not alter the results, similarly to the adjustment for air pollution using a time-varying exposure Cox model [42].

We found that each 100-metre section of the city network between the subject's residence and the nearest public green space increased the risk of death or recurrence by 26%. Our results are consistent with previous demonstrations of reduced mortality among stroke survivors living in greener areas, as reported in Boston [14], Ontario [10] and Catalonia [43]. In addition, three meta-analyses identified a similar effect, but they noted the heterogeneity of the studies and the indicators used [16,44,45].

Our results suggest a differential effect of greenness when recurrence and mortality are considered separately. The beneficial effect of proximity to green spaces from patients' residences was greater and remained significant for stroke

**Table 1. Study population characteristics.**

| Variables | Total subjects n = 360 | Event Composite n = 30[a] | Death n = 15[b] | Recurrence n = 17[b] | Censored n = 330 | P values[c] |
|---|---|---|---|---|---|---|
| **Sex** | | | | | | |
| Men | 160 (44%) | 14 (47%) | 5 (33%) | 10 (59%) | 146 (44%) | 0.80 |
| Women | 200 (56%) | 16 (53%) | 10 (67%) | 7 (41%) | 184 (56%) | |
| **Age** | | | | | | |
| Median age (IQR) | 75 (63-83) | 81 (74-87) | 84 (80-88) | 80 (70-85) | 74 (63-82) | 0.002 |
| 18–64 | 101 (28%) | 4 (13%) | 2 (13%) | 2 (12%) | 97 (29%) | 0.03 |
| 65-79 | 131 (36%) | 9 (30%) | 2 (13%) | 7 (41%) | 122 (37%) | – |
| > 80 | 128 (36%) | 17 (57%) | 11 (74%) | 8 (47%) | 111 (34%) | – |
| **Stroke severity** | | | | | | |
| Median NIHSS score (IQR) | 2 (2-4) | 4 (2-6) | 4 (2-14) | 2 (2-4) | 2 (2-4) | 0.30 |
| - Minor stroke [0–4] | 215 (80%) | 9 (64%) | 4 (57%) | 6 (75%) | 206 (81%) | 0.20 |
| - Moderate to severe stroke [5–42] | 53 (20%) | 5 (36%) | 3 (53%) | 2 (25%) | 48 (19%) | – |
| - NA | 92 | 16 | 8 | 9 | 76 | – |
| Median post-stroke mRS score[d] (IQR) | 1 (0-2) | 2 (1-3) | 2 (1-3) | 1 (0-3) | 1 (0-2) | < 0.001 |
| **Etiological classification of (ischaemic) strokes** | 96 (27%) | 4 (13%) | 3 (20%) | 1 (6%) | 92 (28%) | 0.20 |
| Atheromatous | 75 (21%) | 10 (33%) | 6 (40%) | 4 (24%) | 65 (20%) | – |
| Cardioembolic | 122 (34%) | 9 (30%) | 4 (27%) | 7 (41%) | 113 (34%) | – |
| Lacunar | 67 (18%) | 7 (24%) | 2 (13%) | 5 (29%) | 60 (18%) | – |
| Other | | | | | | |
| **Vascular risk factors** | | | | | | |
| Current hypertension | 229 (64%) | 21 (70%) | 12 (80%) | 10 (59%) | 208 (63%) | 0.40 |
| Current smoking | 93 (28%) | 8 (30%) | 4 (31%) | 5 (31%) | 85 (28%) | 0.80 |
| - NA | 26 | 3 | 2 | 1 | 23 | – |
| Hypercholesterolaemia | 178 (50%) | 14 (47%) | 7 (47%) | 8 (47%) | 164 (50%) | 0.70 |
| - NA | 1 | – | – | – | 1 | – |
| Diabetes | 48 (13%) | 6 (20%) | 5 (33%) | 2 (12%) | 42 (13%) | 0.30 |
| - NA | 1 | – | – | – | 1 | – |
| Atrial fibrillation | 63 (18%) | 10 (33%) | 6 (40%) | 4 (24%) | 53 (16%) | 0.018 |
| - NA | 1 | – | – | – | 1 | – |
| **French deprivation index FDEP** | | | | | | |
| Median decile (IQR) | 7 (5-8) | 7 (5-8) | 8 (6-9) | 7 (5-8) | 6 (5-8) | 0.14 |
| Favoured [1–4] | 73 (32%) | 4 (14%) | 2 (13%) | 3 (18%) | 69 (21%) | 0.30 |
| Mixed [5–7] | 171 (48%) | 13 (43%) | 4 (27%) | 9 (53%) | 158 (48%) | – |
| Unfavoured [8–10] | 116 (32%) | 13 (43%) | 9 (60%) | 5 (29%) | 103 (31%) | – |
| **Public greenness proximity: Mean (sd) Nearest public green space by city network (in metres)** | 309 (192) | 370 (246) | 320 (144) | 400 (300) | 303 (185) | 0.12 |
| **Greenness density: Mean (sd)** | | | | | | |
| **Within 100 metres of residence** | 2.9 (3.3) | 2.7 (4.5) | 4.0 (5.2) | 1.2 (3.1) | 2.9 (3.1) | 0.048 |
| - Area (× 1,000 m²) | 9 (10) | 8 (14) | 13 (17) | 4 (10) | 9 (10) | 0.04 |
| - Proportion area (%) | 5.1 (4.5) | 4.9 (4.8) | 6.7 (5.8) | 2.8 (2.7) | 5.1 (4.4) | 0.50 |
| **Within 400 metres of residence** | 10 (9) | 10 (10) | 13 (11) | 6 (5) | 10 (9) | 0.60 |
| - Area (× 10,000 m²) | | | | | | |
| - Proportion area (%) | | | | | | |
| **Air pollution (µg.m⁻³): Mean (sd)** | | | | | | |
| **$NO_2$ concentrations** | 29.1 (7.5) | 27.6 (7.1) | 25.8 (6.9) | 28.7 (7.0) | 29.2 (7.5) | 0.20 |
| Previous 30 days | 29.3 (7.0) | 27.6 (6.8) | 26.0 (7.3) | 28.5 (6.4) | 29.5 (7.1) | 0.14 |
| Previous 3 months | | | | | | |

*(Continued)*

**Table 1.** (Continued)

| Variables | Total subjects n = 360 | Event | | | Censored n = 330 | P values[c] |
|---|---|---|---|---|---|---|
| | | Composite n = 30[a] | Death n = 15[b] | Recurrence n = 17[b] | | |
| $PM_{10}$ concentrations Previous 30 days Previous 3 months | 18.9 (4.3) 19.1 (3.3) | 19.2 (4.4) 18.8 (3.3) | 19.3 (4.9) 19.3 (4.0) | 18.8 (3.9) 18.1 (2.4) | 18.9 (4.3) 19.1 (3.3) | 0.70 0.40 |

[a] Primary outcome (13 death and 17 recurrences)

[b] Secondary outcomes

[c] Comparison between composite events (primary outcome) and censors

[d] Modified Rankin scale score

IQR: interquartile ranges

sd: standard deviations

recurrence only. This could illustrate a mediating effect of stroke recurrence on the association between greenness and mortality. Moreover, the highest association was noted among the intermediate-aged 65–79-year-old patients. In France, the mean retirement age in 2007 was 61.1 years [46]. A French national study conducted over the same period revealed that the mobility of seniors decreased as their age increased [47]. Several French studies also reported that the percentage of time spent at home was approximately 95% among individuals aged over 65 years, with outdoor activities being restricted to less than 2% of daily life [48,49]. This time drastically decreased from people aged 65–74 years to people aged 75–84 years and people aged > 85 years. Specific data for individuals who had a stroke are not available. However, as we considered patients who returned directly home, *i.e.*, those who had excellent or good immediate post-stroke recovery, as shown by a low post-stroke mRS score, it could be assumed that their lifestyle did not differ greatly from that of the general population. Subjects in the 65–79 years age group could have greater mobility than older subjects did, which could result in more effective access to green spaces. These potential modulations of the greenness effect by age should be considered when comparing results between studies, especially when age structures differ.

Several limitations must be acknowledged. The choice of this five-year period was guided by the need for homogeneity of the environment assessment. This period ended just before the construction of the Dijon tramway network in 2010, which induced major territorial, transportation, and atmospheric changes, limiting the study period to 2009, thus reducing the sample size, the follow-up period, and the number of patients with primary outcomes. Therefore, we cannot rule out the possibility of residual confounding factors. The small number of events in each age group limited additional analyses considering death or recurrence separately across age groups. The data did not allow the determination of the individual access to a private green space; this could have induced misclassification of the subjects having access to a private garden and who were associated with a low value of the greenness density index. The situation is prone to be rare. Finally, the identified relationships could differ across larger cities.

## Perspectives

Larger prospective studies allowing stratified analyses would help to confirmed the results. They will allow exploration of the potential mechanisms involved, such as physical activity, stress reduction, or immune function. If so, it would be necessary to advocate in favour of the greening of urban spaces with decision-makers. Making active use of green spaces a part of prevention should also be included in the recommendations made to professionals in charge of the management of stroke patients. For a clinical perspective, encouraging the use of nearby green spaces may represent a simple, non-pharmacological complement to secondary prevention strategies.

**Table 2. Primary outcome analyses (each greenness index was modelled separately).**

| | Hazard ratio [95% confidence intervals] of green space variables on composite event risk after a first ischaemic stroke Event = death and/or recurrence | | | | |
|---|---|---|---|---|---|
| | Model 1 Crude Composite event (n = 30) | Model 2 Model 1 + sex, age | Model 3 Model 2 + mRS score, FDEP | Model 4 Model 3 + previous 30 day $NO_2$ concentrations | Model 5 Model 3 + previous 30 days $PM_{10}$ concentrations |
| **Public greenness proximity** Nearest public green space by city network [a] | 1.17 [0.99, 1.37] | **1.17 [1.01, 1.37] *** | **1.27 [1.08, 1.49] **** | **1.30 [1.11, 1.53] **** | **1.26 [1.08, 1.48] **** |
| **Greenness density** Area of green space within 100 metres of residence [b] | 0.97 [0.86, 1.09] | 0.97 [0.87, 1.09] | 0.98 [0.87, 1.10] | 0.95 [0.83, 1.07] | 0.96 [0.85, 1.09] |
| Area of green space within 400 metres of residence [c] | 0.99 [0.91, 1.07] | 1.1 [0.91, 1.07] | 0.99 [0.90, 1.09] | 0.95 [0.86, 1.06] | 0.97 [0.88, 1.07] |

[a] Expressed for an increase of 100 metres

[b] Expressed for an increase of 1,000 m²

[c] Expressed for an increase of 10,000 m²

FDEP: French deprivation index, mRS: modified Rankin scale

Values in bold mean that they are statistically significant

* P < 0.05

** P < 0.01

**Table 3. Secondary outcomes analyses (each greenness index was modelled separately).**

| | Death after a first ischaemic stroke Hazard ratio [95% confidence interval] of green space variables | | | Recurrence after a first ischaemic stroke Hazard ratio [95% confidence interval] of green space variables | | |
|---|---|---|---|---|---|---|
| | Model A-1 Crude Event: death (n = 15) | Model A-2 Model A-1 + sex, age, mRS score, FDEP | Model A-3 Model A-2 + previous 30 days $PM_{10}$ concentrations | Model B-1 Crude Event: recurrence (n = 17) | Model B-2 Model B-1 + sex, age, mRS score, FDEP | Model B-3 Model B-2 + previous 30 days $PM_{10}$ concentrations |
| **Public greenness proximity** Nearest public green space by city network [a] | 1.04 [0.80, 1.36] | 1.17 [0.90, 1.51] | 1.17 [0.89, 1.52] | **1.23 [1.02, 1.50] *** | **1.29 [1.07, 1.56] **** | **1.30 [1.07, 1.58] **** |
| **Greenness density** Area of green space within 100 metres of residence [b] | 1.08 [0.96, 1.23] | 1.09 [0.96, 1.25] | 1.07 [0.93, 1.22] | **0.73 [0.54, 0.97] *** | **0.72 [0.54, 0.97] *** | **0.71 [0.53, 0.96] *** |
| Area of green space within 400 metres of residence [c] | 1.06 [0.97, 1.16] | 1.10 [0.98, 1.22] | 1.06 [0.95, 1.19] | **0.80 [0.66, 0.98] *** | **0.77 [0.61, 0.97] *** | **0.76 [0.60, 0.96] *** |

[a] Expressed for an increase of 100 metres

[b] Expressed for an increase of 1,000 m²

[c] Expressed for an increase of 10,000 m²

FDEP: French deprivation index, mRS: modified Rankin scale

Values in bold mean that they are statistically significant

* P < 0.05 ** P < 0.01

To conclude, this study highlighted the beneficial influence of greenness on post-stroke outcomes in an urban area. The proximity to urban public green spaces from the place of residence and greenness density were associated with a reduced risk of recurrence after ischaemic stroke. These results indicate that in addition to medical strategies, urban

**Table 4. Age groups sensitivity analyses (each greenness index was modelled separately).**

| | By age groups, HR [95% CI] of green space variables on composite event risk after a first ischaemic stroke Event = death and/or recurrence Models adjusted for sex, mRS score, FDEP, previous 30 days PM$_{10}$ concentrations | | |
|---|---|---|---|
| | ≤ 64 years (n event = 4) | 65–79 years (n event = 9) | ≥ 80 years (n event = 17) |
| **Public greenness proximity** Nearest public green space by city network [a] | 1.12 [0.66, 1.88] | **1.37 [1.10, 1.71] \*\*** | 1.18 [0.92, 1.50] |
| **Greenness density** Area of green space within 100 metres of residence [b] | 1.16 [0.92, 1.48] | 0.92 [0.70, 1.21] | 0.92 [0.77, 1.09] |
| Area of green space within 400 metres of residence [c] | 1.11 [0.95, 1.31] | 0.98 [0.80, 1.19] | 0.87 [0.73, 1.04] |

CI: Confidence IntervalHR: hazard ratioFDEP: French deprivation indexmRS: modified Rankin scale

[a] Expressed for an increase of 100 metres

[b] Expressed for an increase of 1,000 m²

[c] Expressed for an increase of 10,000 m²

Value in bold means that it is statistically significant

\*\* $P < 0.01$

planning policies, secondary prevention based on the use of nearby greenness could positively impact patients after stroke.

## Acknowledgments

The authors acknowledge Mathieu Boilleaut from the local association for air quality monitoring Atmo Bourgogne Franche-Comté, Bérenger Martin and Clémence Courbot from CHU Besançon, Damien Roy from ThéMA laboratory. Computations for environmental exposure have been performed on the supercomputer facilities of the *Mésocentre de calcul de Franche-Comté*.

## Author contributions

**Conceptualization:** Nadine Bernard, Anne-Sophie Mariet, Anne-Laure Parmentier, Frédéric Mauny.

**Formal analysis:** Raphaël Anxionnat, Kadiatou Diallo.

**Funding acquisition:** Nadine Bernard, Sophie Pujol, Yannick Bejot, Frédéric Mauny.

**Investigation:** Sophie Pujol, Helene Houot, Yannick Bejot.

**Methodology:** Anne-Sophie Mariet, Sophie Pujol, Anne-Laure Parmentier, Frédéric Mauny.

**Supervision:** Nadine Bernard, Yannick Bejot, Frédéric Mauny.

**Validation:** Nadine Bernard, Sophie Pujol, Frédéric Mauny.

**Writing – original draft:** Raphaël Anxionnat.

**Writing – review & editing:** Nadine Bernard, Anne-Sophie Mariet, Sophie Pujol, Theophile Pierre, Yannick Bejot, Frédéric Mauny.

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
