## [Decision Letter · Decision Letter 0]

6 Aug 2025

PONE-D-24-52535Benefit of urban greenness on patients after an ischaemic stroke: mortality or recurrence? A registry-based cohort studyPLOS ONE?

Dear Dr. Mauny,

Thank you for submitting your manuscript to PLOS ONE. After careful consideration, we feel that it has merit but does not fully meet PLOS ONE’s publication criteria as it currently stands. Therefore, we invite you to submit a revised version of the manuscript that addresses the points raised during the review process.

We look forward to receiving your revised manuscript.

Kind regards,

Dr Redoy Ranjan, MBBS, MRCSEd, Ch.M., MS (CV&TS), FACS

Academic Editor

PLOS ONE

Journal Requirements:

5. In the online submission form, you indicated that “Data cannot be shared publicly because of Europrean LEgacy restriction. Data are available from the CHU of Besancon (contact via the coresponding author) for researchers who meet the criteria for access to confidential data.”

6. Please include captions for your Supporting Information files at the end of your manuscript, and update any in-text citations to match accordingly. Please see our Supporting Information guidelines for more information: http://journals.plos.org/plosone/s/supporting-information .

7. Please remove all personal information, ensure that the data shared are in accordance with participant consent, and re-upload a fully anonymized data set.

Additional guidance on preparing raw data for publication can be found in our Data Policy (https://journals.plos.org/plosone/s/data-availability#loc-human-research-participant-data-and-other-sensitive-data) and in the following article: http://www.bmj.com/content/340/bmj.c181.long .

8. Thank you for stating the following financial disclosure:

“NO2 pollution exposure assessment was supported by the Agence de l'Environnement et de la Maîtrise de l'Énergie (ADEME) [1217C0065] as part of the National Environmental Research and Occupational Health Program 2012 from the French Agency for Food, Environmental and Occupational Health and Safety (ANSES). PM10 pollution exposure assessment was supported by the Fondation de France

(Engagement 00089823). The Dijon Stroke Registry was funded by Santé Publique France (French Institute for Public Health Surveillance), Institut national de la santé et de la recherche médicale (INSERM), and CHU Dijon Bourgogne.”

9. Thank you for stating the following in the Competing Interests section:

“NO2 pollution exposure assessment was supported by the Agence de l'Environnement et de la Maîtrise de l'Énergie (ADEME) [1217C0065] as part of the National Environmental Research and Occupational Health Program 2012 from the French Agency for Food, Environmental and Occupational Health and Safety (ANSES). PM10 pollution exposure assessment was supported by the Fondation de France

(Engagement 00089823). The Dijon Stroke Registry was funded by Santé Publique France (French Institute for Public Health Surveillance), Institut national de la santé et de la recherche médicale (INSERM), and CHU Dijon Bourgogne.”

We note that you received funding from a commercial source: BMS, Pfizer, Medtronic, Amgen, Servier,

NovoNordisk, Novartis

Additional Editor Comments:

The authors are thanked for this submission to PLOS ONE. After a critical external peer review by three experts and considering the overall reviewers' comments, I recommend improving your paper's clarity and presentation based on recent literature and acknowledging reviewers' concerns. PLOS ONE's publication criteria consider methodological rigour and ethical standards, regardless of the paper's novelty.

Reviewer's Responses to Questions

**Comments to the Author**

1. Is the manuscript technically sound, and do the data support the conclusions?

Reviewer #1: Yes

Reviewer #2: Yes

2. Has the statistical analysis been performed appropriately and rigorously?

Reviewer #1: Yes

Reviewer #2: Yes

3. Have the authors made all data underlying the findings in their manuscript fully available?

Reviewer #1: Yes

Reviewer #2: Yes

4. Is the manuscript presented in an intelligible fashion and written in standard English?

Reviewer #1: Yes

Reviewer #2: Yes

Reviewer #1: Reviewer comments for PLOS ONE (PONE-D-24-52535):

The authors demonstrated the beneficial effects of greenness on post-ischemic stroke recurrence patients in an urban area using a registry based cohort study. However, owing to the limited no. of a large amount of information such as theoretical, logically consistent (rational and research studies), statistical analysis, and survey database information does not get into my head (especially in the introduction, results, and discussion section). Based on the above limitation, I do have some major and minor concerns that should be addressed by the authors.

Major points-

1). Please add appropriate keywords related to this study and check keywords with abstract section content.

2). Introduction section is too short and needs to be elaborated.

3) Clearly state the research gap or knowledge deficit in the introduction section that the current study aims to address. Emphasize why investigating relationship between urban greeness and ischemic stroke patients is important and how it contributes to filling this gap in the literature.

4). Authors should maintain consistency, if once used abbreviation in the manuscript.

5). While conclusion section covers various aspects related to the beneficial effects of greenness on post-ischemic stroke recurrence patients in an urban area using a registry based cohort study, however authors should add as a suggestion, future research directions and clinical implications based on the findings of the current study.

6). Discussion section is too large. It is highly encouraged that the author should includes only relevant things with justify to results and also articles in discussion that can directly be correlated with the current study. Try to concise the discussion section.

Reviewer #2: As the age of strokes is getting older and stroke severity is becoming milder, research on recurrence after stroke is valuable. The authors' study showed the beneficial influence of greenness on post-stroke outcomes in an urban area, suggesting the importance of the living environment after stroke.

To complete this paper, I would like to offer some suggestions.

Regarding the last line on page 7, "The area and percentage of green spaces within radii of 100 and 400 metres around the 7places of residence were calculated for each patient," why was "The area and percentage of green spaces" decided to be "within radii of 100 and 400 metres"? It would be better to describe the reason why to deside "within radii of 100 and 400 metres".

Regarding the line 3-5 on page 10, "After one year of follow-up, there were 30 composite events (13 deaths and 17 recurrences). Regarding separate death and recurrence events, there were 15 and 17 deaths and recurrences," , were 2 of the 17 recurrences deaths?

Regarding lines 7-8 of the Results on page 10 “Patients who presented a subsequent event (death or recurrence) were older and had higher post-stroke mRS scores.”, I am unable to find data on higher post-stroke mRS scores. While this may be an oversight on my part, if the data is not listed, the authors should include it in the Results section.

**Do you want your identity to be public for this peer review?** For information about this choice, including consent withdrawal, please see our Privacy Policy

Reviewer #1: **Yes:** HITESH S CHAOUHAN

Reviewer #2: No

---

## [Author Response · Author response to Decision Letter 1]

12 Nov 2025

5. Review Comments to the Author

Reviewer #1: Reviewer comments for PLOS ONE (PONE-D-24-52535):

The authors demonstrated the beneficial effects of greenness on post-ischemic stroke recurrence patients in an urban area using a registry based cohort study. However, owing to the limited no. of a large amount of information such as theoretical, logically consistent (rational and research studies), statistical analysis, and survey database information does not get into my head (especially in the introduction, results, and discussion section). Based on the above limitation, I do have some major and minor concerns that should be addressed by the authors.

Major points-

1). Please add appropriate keywords related to this study and check keywords with abstract section content.

We removed the less relevant keywords and added one, all of which are present in the study abstract. Please see page 4. The list of the keywords includes : stroke, environmental health, cohort studies, greenness, population-based study. We deleted “neurology “ and “epidemiology”.

2). Introduction section is too short and needs to be elaborated.

In response to both point 2 and point 3, and as also requested by the second reviewer, we totally modified and expanded the Introduction part to address the importance of studying the outcomes of stroke survivors, the relevance of investigating the impact of greenness on stroke recurrence risk, and the pathophysiological mechanisms underlying the presumed effects. Please see page 4 to page 6.

3) Clearly state the research gap or knowledge deficit in the introduction section that the current study aims to address. Emphasize why investigating relationship between urban greenness and ischemic stroke patients is important and how it contributes to filling this gap in the literature.

We clarified these elements in the introduction section, as indicated in point 2 above.

4). Authors should maintain consistency, if once used abbreviation in the manuscript.

We revised the manuscript to ensure consistent use of abbreviations, for example “mRS” (modified Rankin Scale or post-stroke Rankin Scale), and removed unnecessary abbreviations to improve clarity.

5). While conclusion section covers various aspects related to the beneficial effects of greenness on post-ischemic stroke recurrence patients in an urban area using a registry based cohort study, however authors should add as a suggestion, future research directions and clinical implications based on the findings of the current study.

In response, we revised the conclusion section to add suggestions for future research directions and to highlight the potential clinical implications of our findings, including the encouragement of nearby green space use as part of secondary prevention strategies. The manuscript was modified as follows: page 14, Perspective part, lines 2-3 “They will enable exploration of the potential mechanisms involved, such as physical activity, stress reduction, or immune function.” and page 14-15, last two lines of the Perspective part “For a clinical perspective, encouraging the use of nearby green spaces may represent a simple, non-pharmacological complement to secondary prevention strategies.”

6). Discussion section is too large. It is highly encouraged that the author should include only relevant things with justify to results and also articles in discussion that can directly be correlated with the current study. Try to concise the discussion section.

The authors thanks for this comment. Indeed, the discussion part was not focused on the main topic of the results. We reduced the length of the discussion (from 82 lines to 61 lines). In response, and also to address points 2 and 3, we moved the discussion on the pathophysiological mechanisms to the Introduction, and reduced the sections on methodological biases, comparisons with studies reporting similar results, and explanations related to the study period. Please see page 12 to page 14.

Reviewer #2:

As the age of strokes is getting older and stroke severity is becoming milder, research on recurrence after stroke is valuable. The authors' study showed the beneficial influence of greenness on post-stroke outcomes in an urban area, suggesting the importance of the living environment after stroke. To complete this paper, I would like to offer some suggestions.

Regarding the last line on page 7, "The area and percentage of green spaces within radii of 100 and 400 metres around the places of residence were calculated for each patient," why was "The area and percentage of green spaces" decided to be "within radii of 100 and 400 metres"? It would be better to describe the reason why to deside "within radii of 100 and 400 metres".

In response, we revised the Methods section to clarify the rationale for defining greenness within 100- and 400-m radii. We provided more information and references considering the way the areas were defined, and explicated the living and the walking neighborhoods. We also inserted bibliographic references supporting this approach. The method part was modified as follows, page 8: ”This index was computed for the area of the walking and living neighborhood (ref Smith et al 2010; Choix et an 2009). For each patient, we defined these areas using a circular buffer around the residence. According to Forsyth et al (2008) and our previous studies (ref Tenailleau et al,2014 et 2015), the area and percentage of green spaces were defined calculated within radii of 100 and 400 metres.

Regarding the line 3-5 on page 10, "After one year of follow-up, there were 30 composite events (13 deaths and 17 recurrences). Regarding separate death and recurrence events, there were 15 and 17 deaths and recurrences," , were 2 of the 17 recurrences deaths?

We clarified this point at the beginning of the results section. Please see page 10, third line of the result part: ” After one year of follow-up, 30 subjects experienced a composite event (13 deaths and 17 recurrences). Among these subjects, two experienced a recurrence and died before the end of the one-year follow up”.

Regarding lines 7-8 of the Results on page 10 “Patients who presented a subsequent event (death or recurrence) were older and had higher post-stroke mRS scores.”, I am unable to find data on higher post-stroke mRS scores. While this may be an oversight on my part, if the data is not listed, the authors should include it in the Results section.

The information was not sufficiently clear; we have specified the values in the results section and renamed the variable as “Median post-stroke mRS score” in Table 1 (study population characteristics). Please see page 11, lines 2 to 5: “Patients who presented a subsequent event (death or recurrence) were older (median age 81 years [IQR 74–87] vs. 74 years [IQR 63–82], p = 0.002) and had higher post-stroke mRS scores (median 2 [IQR 1–3] vs. 1 [IQR 0–2], p < 0.001).”

---

## [Decision Letter · Decision Letter 1]

3 Dec 2025

Benefit of urban greenness on patients after an ischaemic stroke: mortality or recurrence? A registry-based cohort study

PONE-D-24-52535R1

Dear Dr. Mauny,

We’re pleased to inform you that your manuscript has been judged scientifically suitable for publication and will be formally accepted for publication once it meets all outstanding technical requirements.

Kind regards,

Redoy Ranjan, MS (CV&TS), Ch.M. (Edin), PhD

Academic Editor

PLOS ONE

Additional Editor Comments (optional):

Reviewers' comments:

Reviewer's Responses to Questions

**Comments to the Author**

Reviewer #2: (No Response)

2. Is the manuscript technically sound, and do the data support the conclusions?

Reviewer #2: Yes

3. Has the statistical analysis been performed appropriately and rigorously?

Reviewer #2: Yes

4. Have the authors made all data underlying the findings in their manuscript fully available?

Reviewer #2: Yes

5. Is the manuscript presented in an intelligible fashion and written in standard English?

Reviewer #2: Yes

Reviewer #2: The authors have considered the comments I made in my review and have revised the manuscript appropriately.

**Do you want your identity to be public for this peer review?** For information about this choice, including consent withdrawal, please see our Privacy Policy

Reviewer #2: No

---

## [Editor Report · Acceptance letter]

PONE-D-24-52535R1

PLOS One

Dear Dr. Mauny,

I'm pleased to inform you that your manuscript has been deemed suitable for publication in PLOS One. Congratulations! Your manuscript is now being handed over to our production team.

Kind regards,

on behalf of

Dr. Redoy Ranjan

Academic Editor

PLOS One